# Comparison between Self-Completed and Interviewer-Administered 24-Hour Dietary Recalls in Cancer Survivors: Sampling Bias and Differential Reporting

**DOI:** 10.3390/nu14245236

**Published:** 2022-12-08

**Authors:** Rana Conway, Gabriella Heuchan, Helen Croker, Sara Esser, Victoria Ireland, Phillippa Lally, Rebecca Beeken, Abigail Fisher

**Affiliations:** 1Department of Behavioural Science and Health, University College London, London WC1E 6BT, UK; 2Leeds Institute of Health Sciences, Leeds LS2 9JT, UK

**Keywords:** dietary assessment, diet recall, diet, 24 hour recall, nutrient intake, cancer

## Abstract

Self-completed 24 h dietary recalls (24-HRs) are increasingly used for research and national dietary surveillance. It is unclear how difficulties with self-completion affect response rates and sample characteristics. This study identified factors associated with being unable to self-complete an online 24-HR but willing to do so with an interviewer. Baseline 24-HRs from the ASCOT Trial were analysed (*n* = 1224). Adults who had been diagnosed with cancer in the past seven years and completed treatment, were invited to self-complete 24-HRs online using myfood24^®^. Non-completers were offered an interviewer-administered 24-HR. One third of participants willing to provide dietary data, were unable to self-complete a 24-HR. This was associated with being older, non-white and not educated to degree level. Compared to interviewer-administered 24-HRs, self-completed 24-HRs included 25% fewer items and reported lower intakes of energy, fat, saturated fat and sugar. This study highlights how collection of dietary data via online self-completed 24-HRs, without the provision of an alternative method, contributes to sampling bias. As dietary surveys are used for service and policy planning it is essential to widen inclusion. Optimisation of 24-HR tools might increase usability but interviewer-administered 24-HRs may be the only suitable option for some individuals.

## 1. Introduction

Dietary assessment is an essential element of many epidemiology studies and dietary interventions. Yet, achieving accurate dietary assessment with limited resources remains a challenge for researchers. The most used tools, including diet diaries, 24 h dietary recalls (24-HRs) and food frequency questionnaires (FFQs), come with well recognized strengths and limitations [1,2]. Biomarkers may be seen as the ideal surrogate measure, free from the problems associated with self-reporting; however, biomarkers do not solely reflect intake and divorcing nutrients from foods is not helpful when providing individual recommendations nor from a policy perspective [3].

24-HRs appear to enable more accurate assessments of short-term energy and protein intake than FFQs and are increasingly favored for epidemiology [4]. Interviewer-administered 24-HRs present considerable researcher and respondent burden but self-completed online 24-HRs offer an attractive alternative as they can be completed at a time convenient to respondents, coding is integrated within the collection process and they reduce researcher burden, and therefore costs, considerably [2,5,6]. Tools generally involve respondents entering search terms for foods and drinks consumed and then selecting appropriate items from drop down lists; with portion sizes estimated using food photographs and prompts provided for items commonly eaten together [7]. Several groups have compared intakes recorded using self-completed 24-HRs with interviewer-administered 24-HRs and found that on average, underreporting may be higher with the former, but given the many advantages they have generally concluded that they perform favorably and provide a good alternative [8,9,10,11]. A smaller number of research groups have carried out validation studies against objective measures of intake, such as direct observation of intake or biomarkers. These have generally found underreporting with self-completed 24-HRs that is broadly in line with other dietary assessment methods, with some variation between different systems. Energy intake has been reported to be underestimated by 32% with myfood24^®^ (Dietary Assessment Ltd., Leeds, UK) [12], 25% with Intake24^®^ (University of Cambridge, Cambridge, UK) [9] and 17% ASA24^®^ (National Cancer Institute, Rockville, MD, USA) [13], compared to biomarkers.

Some level of support or researcher contact is generally needed with self-completion of 24-HRs, and they may not suit all populations which could disproportionately reduce response rates [14]. While adolescents may seem particularly likely to embrace technology, one study found participants aged 12–17 years old took an average of 25 min to self-complete recalls, compared to 15 min for interviewer-administered recalls, and they expressed a preference for the latter [15]. Tailored support may increase response rates [14,16]. A Canadian study assessing usability of the ASA24^®^-Canada (National Cancer Institute, Rockville, MD, USA) tool with 17 participants on a low-income found only one successfully completed their 24-HR unassisted [17]. A UK-based study with myfood24^®^ found older adults were less likely to submit self-completed 24-HRs, as were individuals with a lower self-rating of internet ability and less frequent internet use [18]. Problems with usability were also demonstrated among sixty older adults using the Norwegian version of myfood24^®^, with only 16 individuals assessing the usability of the tool as satisfactory [19]. As we increasingly move to self-completed 24-HRs for intervention studies and national surveys, such as the UK’s National Diet and Nutrition Survey (NDNS) [20], it is important to understand whether, and in what way, responses might be affected. It may be necessary to include an alternative method of dietary assessment to increase response rates from some sub-groups of the population.

The aims of the current study were to (1) identify factors associated with being unable to self-complete a 24-HR but able to do so with an interviewer, and (2) explore how dietary data collected by self-completed 24-HR compared to interviewer-administered 24-HRs, in cancer survivors.

## 2. Methods

### 2.1. Study Participants

A cross-sectional study was carried out using baseline data from the Advancing Survival Cancer Outcomes Trial (ASCOT) [21]. Study procedures and sample size are described elsewhere [21]. In brief, ten hospital sites across London and Essex were asked to send surveys to patients aged 18 years and older diagnosed with breast, prostate or colorectal cancer in the past 4 years, between 2012 and 2015. Respondents were asked to provide contact details if interested in learning more about a trial of a lifestyle intervention (ASCOT). Those expressing interest, meeting ASCOT eligibility criteria outlined above, and providing informed consent were asked to complete additional baseline assessments including two 24-HRs between 2015 and 2019. The current analysis includes eligible participants randomized within the trial, who completed the first 24-HR requested at baseline.

### 2.2. Survey

Participants were sent a paper survey and a link to an online version of the same survey and invited to choose how they completed and returned this (postal or online). The survey included questions to assess age (in years), gender (male or female), ethnicity (16 subcategories were presented but for the current analysis this was dichotomized into white and non-white) and cancer type (breast, prostate, or colorectal cancer). Highest level of education was assessed by asking about educational/professional qualifications and categorizing responses into no formal qualifications, General Certificate of Secondary Education (GCSE)/vocational, A-level or equivalent, degree or higher. Marital status was assessed by asking about 5 categories, which were aggregated into 3 categories (married/cohabiting, separated/divorced/widowed or single). To determine employment situation 9 categories were presented, which were aggregated into 3 broad categories (employed/self-employed, retired or other). Body Mass Index (BMI) was calculated using self-reported height and weight and was categorized as underweight/normal weight (<25), overweight (≥25 and <30), and obese (≥30) [22]. The underweight and normal weight categories were combined as less than 0.5% of participants fell into the underweight category (BMI < 18.5). Index of Multiple Deprivation (IMD) decile was calculated based on the participant’s post code, with lower IMD scores representing higher levels of deprivation [23]. Participants were also asked ‘Do you use the internet? (e.g., for health information)’ and ‘Do you use email?’, with response options yes or no.

### 2.3. Collection of 24-HRs

Letters were sent to participants informing them of the dates they were due to complete 24-HRs, which were scheduled to collect dietary data for one weekday and one weekend day. On the allotted date for the first 24-HR, which was a weekday, all participants supplying an email address were sent an email invitation with instructions for self-completing their 24-HR and a personalized electronic link to myfood24^®^ [24]. Participants who did not use email were contacted by telephone on the allotted day and a researcher collected dietary information and entered this directly into myfood24^®^. Participants without an email address were therefore automatically allocated to interviewer-administered 24-HRs. Details about dietary supplements were collected at the end of the 24-HR but were not included in the current analysis. Participants were asked to telephone or email the research team if they had any questions or problems completing the 24-HR. Researchers phoned participants who had not self-completed their 24-HR as a reminder and to offer support. Some participants not initially submitting a 24-HR were able to do so with minimal support, for example a new electronic link or reassurance that exact brand matching was unnecessary. Several participants reported encountering problems with self-completion, including lacking technological skills, being unable to find foods and being overwhelmed by long lists of foods. If participants described making several attempts but experiencing problems with completing the recall, they were given the option of completing it via telephone. Participants were therefore defined as being ‘unable’ to self-complete a 24-HR if they did not have an email address or if they reported experiencing considerable problems when attempting self-completion. Self-completed 24-HRs were not routinely reviewed at the time of submission but if researchers noticed 24-HRs that looked problematic they phoned participants to resolve issues or offer the interviewer-completion alternative. For example, a recall containing just 5 items was noticed by a researcher, including 1 g chapati flour and 1 g of fried aubergine. In this case, the participant reported eating one chapati and one fried egg and had selected the first items appearing in the lists presented to them after entering the search terms ‘chapati’ and ‘fried egg’. This participant reported that no results were found when they searched for other items consumed, so they had omitted these from the original 24-HR submission.

### 2.4. Processing 24-HR Data

Information collected from the first of the two 24-HRs was examined for this study. Data were exported from myfood24^®^ as an Excel file with each food or drink item on a separate row alongside portion and nutrient information, and multiple rows per participant. The method of recall completion was not included in this data file. The dataset was then examined by experienced researchers who were Registered Dietitians or individuals with a post-graduate qualification in nutrition to identify possible data entry errors. Food items, excluding drinks, were first sorted in descending order according to size of portion eaten. A researcher then visually examined the data and flagged any items that appeared much larger or smaller than might be expected. As this was an exploratory exercise no cut-off values were set and the researcher took a pragmatic approach, for example flagging 500 g of cheese as particularly large, but not 500 g of spaghetti Bolognese. Likewise, flagging 1 g of bread as a particularly small portion, but disregarding 1 g of dried herbs.

Food-level nutrient data was aggregated to produce estimates of total nutrient intake and 24-HRs with particularly high daily totals for energy, sugar, fat or fibre were flagged. All 24-HRs flagged for containing possible errors at the food level, or total nutrient per day level, were examined more closely to consider eating occasion, context and portion options presented in myfood24^®^. When one researcher believed they had identified a true error this was only changed if a second researcher agreed the entry was highly implausible, for example 4000 g of watercress was changed to 40 g of watercress. A record was kept of all unusually large or small portions identified, probable selection errors identified and any changes made to the 24-HR.

Once 24-HR data had been examined and any changes had been made, data were re-extracted from myfood24^®^. Percentage energy from each macronutrient was calculated using the factors 3.75 kcal/g carbohydrate, 4 kcal/g protein and 9 kcal/g fat. As this was exploratory analysis all 24-HRs submitted were included.

### 2.5. Statistical Analysis

Statistical analysis was carried out using the Statistical Package for the Social Sciences (SPSS) version 25 (IBM Corp., Armonk, NY, USA). *t*-tests and Chi-squared tests were used to descriptively explore differences in demographics and technology usage between individuals self-completing 24-HRs and those requiring interviewer-administered 24-HRs.

Logistic regression analyses were conducted to explore factors associated with method of completing 24-HR, with interviewer-administered 24-HR as the target variable (independent variable) to allow factors associated with being unable to self-complete a 24-HR, to be highlighted. First a series of regressions were run for each of the independent variables, with no covariates in the model. Then, one logistic regression analysis was run to assess associations between demographic characteristics (age, gender, ethnicity, highest level of education, marital status, employment, IMD) and BMI category and the dichotomous dependent variable (interviewer-administered or self-completed 24-HR). Reference categories were male, white, degree or higher, married/co-habiting, employed and underweight/healthy weight. Cancer type was not included as a covariate to avoid the problem of multicollinearity, as cancer type and gender were identical in the breast cancer sample (all female) and prostate cancer sample (all male). Internet use and email use were also excluded as they are likely mediators of association between demographic characteristics and ability to self-complete a 24-HR. IMD decile was treated as a continuous variable as it is a rank-based indice [25]. Missing value analysis was conducted, and multiple imputation was used with 5 iterations to account for all variables included in the regression analysis [26]. This imputation was repeated to check whether results were similar. Missing value analysis found that 1.8% of 13,464 values were missing and 16.8% of 1224 cases had at least 1 piece of missing data. The final logistic regression was also repeated with the completers sample to explore if similar results were achieved.

Independent samples t-tests were used to compare nutrient intake estimates from self-completed and interviewer-administered 24-HRs and chi-square was used to compare prevalence of unusual portions sizes and probable selection errors.

## 3. Results

In total, 1229/1348 (91%) ASCOT participants completed the first baseline 24-HR. Data from five participants were excluded from the current analysis as their 24-HRs had been completed via a combination of phone, email and post. The first recall was scheduled for a weekday but was sometimes completed on a weekend day. Of the 1224 participants included in this analysis, two thirds (*n* = 809) were able to self-complete a 24-HR but a third (*n* = 415) required an interviewer-administered 24-HR (Table 1). The mean age of participants self-completing recalls was 61.6 years, compared to 69.8 years for those requiring an interviewer-administered 24-HR. Although more internet and email users were able to self-complete the 24-HR, one in five participants (21.7%) describing themselves as both internet and email users were unable to do so.

Table 2 shows the unadjusted and adjusted results from the logistic regression analyses conducted with the imputed dataset. The adjusted analysis showed people were less likely to be able to self-complete a 24-HR using myfood24^®^ if they were older, non-white and not educated to degree level. Repeating this analysis with a second imputed dataset, and with the dataset only including completers, showed the same associations (see Appendix A).

Self-completed 24-HRs included 25% fewer food and drink items per day compared to interviewer-administered 24-HRs (mean 17.6 vs. 23.9 items) (Table 3). The mean energy intake estimated from self-completed 24-HRs was 106 calories fewer than interviewer-administered 24-HRs and estimates of absolute intake of all macronutrients were lower. Percentage calories from protein and total carbohydrates was not associated with method of 24-HR completion but percentage calories from sugar, fat and saturated fat was lower for self-completed 24-HRs.

Unusually large portions, small portions and probable selection errors were identified more often in self-completed 24-HRs (Table 4).

## 4. Discussion

One third of ASCOT participants willing to provide dietary data, were unable to self-complete a 24-HR using myfood24^®^. Requiring an interviewer-administered 24-HR instead, was associated with being older, non-white and not being educated to university degree level. Self-completed 24-HRs included 25% fewer items, reported lower intakes of energy and lower intakes of fat, saturated fat and sugar as a percentage of energy, as well as a greater number of entries thought to be errors.

Most epidemiological studies use a single method of dietary assessment, with self-completed 24-HRs increasingly being the method of choice. Some non-completion is inevitable irrespective of research method, but these results suggest choice of dietary assessment methodology may disproportionately reduce response rates and alter the results. Despite attempting to contact all ASCOT participants not self-completing 24-HRs, nearly one in ten still failed to complete a 24-HR. The completion rate of 91% was higher than among a comparable population of older adults in the UK taking part in the CHARIOT-Pro study, where 67% of participants self-completed a 24-HR, despite CHARIOT-Pro participants receiving a demonstration of the use of myfood24^®^ [27]. Providing a helpline and timely support is good practice, but individuals may be reluctant to ask for help. Very few participants in ASCOT phoned for assistance, and if researchers had not initiated contact, it is likely that dietary intake would not have been assessed for a larger number of participants. Results of a feasibility study for the Alberta’s Tomorrow Project suggest that even when participants call a helpline and receive support, they do not always go on to successfully self-complete a 24-HR [16]. The feasibility study found a third of 331 consented participants, did not complete any of the four 24-HRs requested, although half of these had called the help desk at least once and 10% had made three or more contacts [16]. ASCOT participant’s reports of problems using computers generally, echo those of older adults in other studies [14,27]. In large surveys, a telephone help line or other methods of support may not be provided, which could exacerbate sampling bias and problems in self-completing 24-HRs. Some ASCOT participants also mentioned feeling overwhelmed having to scroll through long lists of branded items. While having a limited number of items in a food database may be seen as problematic [28], users of myfood24^®^-Germany also criticized the high number of search results [29] and young adults evaluating Myfood24^®^-Germany reported similar issues, despite one fifth having a nutrition background [29]. The Irish 24-HR tool Foodbook24 (University College Dublin, Dublin, Ireland) and the UK’s Intake24^®^ system deliberately contain a limited number of items to reduce participant burden as well as to allow more efficient management of updates [20,30].

Analysis of demographic characteristics showed individuals belonging to groups often under-served in health research, were more likely to struggle with self-completing a 24-HR. As the population ages in many countries, monitoring of dietary intake and intervention delivery will increasingly be required for older adults. Protocols including well designed initial training sessions or instructional videos may be helpful and in time, older adults may be more experienced with technology, but this would need to be assessed [1]. Overrepresentation of participants who are highly educated, of higher socio-economic status and white is a limitation of much research and choice of dietary assessment method may exacerbate this problem and contribute to sampling bias. This has far-reaching implications as population level dietary data is used when planning service provision and national food policy. For adults without formal educational qualifications, the odds of being unable to self-complete a 24-HR are 4.29 (95% confidence interval (CI) 2.87–6.40) times those of individuals with a degree (Table 2). According to the National Literacy Trust, 16% of adults in England can be described as having ‘very poor literacy skills’, and these individuals would likely be excluded from surveys requiring self-completion of 24-HRs which involves text heavy instructions and pages of food options [31]. A Canadian study assessing the usability of ASA24^®^-Canada with participants with a low-income, using qualitative methods including a think-aloud procedure, revealed problems such as their search terms not leading to the appropriate item in the database, misunderstanding questions and terminology not being understood [17]. Optimizing 24-HR systems to recognize a higher number of misspellings and returning shorter lists of options would help those with lower literacy levels but may not be sufficient to allow all willing individuals to participate. Non-white ASCOT participants were more likely than white participants to switch to interviewer-administered 24-HR (odds ratio (OR) 3.43, CI 1.99–5.91), which is unsurprising as ASCOT researchers also struggled to find suitable matches for a range of foods eaten more often by individuals from minority ethnic groups, such as goat curry, jollof rice and barfi. Problems relating to a lack of appropriate foods and portion estimation tools to enable accurate dietary assessment in minority ethnic groups is increasingly being recognized as a challenge which needs to be addressed to reduce sampling bias and facilitate more accurate dietary assessment [32,33].

While we found self-completed 24-HRs included 25% fewer items, Public Health England found the number of items recorded in Intake24 was broadly similar to that recorded in paper diaries used in previous years for the National Diet and Nutrition Survey [20]. Similarly, a pilot study for ASA24^®^-Kids-2014 (National Cancer Institute, Rockville, MD, USA) found adolescents reported a similar number of items when randomly assigned to self-completion or interviewer administered 24-HR [15]. This could be due to the different characteristics of the two groups but other factors may also explain the discrepancy. Researchers may have been more likely than participants to enter composite foods as their constituent parts and enter several items to approximate an equivalent nutrient profile to missing items. Interviewers also provided more prompts than myfood24^®^ for items commonly eaten together and during interviews participants were unable to skip the list presented at the end of the 24-HR of commonly forgotten items, such as snacks. This is supported by the finding that percentage of calories from sugar, fat and saturated fat were lower in self-completed 24-HRs. A comparison between self-completed and interviewer-administered 24-HRs in Washington found 12.6% of sweets, snacks and desserts were omitted with the former and 2.5% with the latter, compared to true intake observed at a buffet [11]. We might expect individuals to be more willing to report items perceived as unhealthy to an interviewer, compared to a computer, but this would suggest otherwise. Our finding regarding lower total nutrient intakes recorded with self-completed 24-HRs is consistent with previous findings of around 10–20% lower estimates, compared with those derived from interviewer-administered 24-HRs [12]. As systematic differences in nutrient intake are evident between the two methods of assessment, if both were used this would need to be considered when exploring associations between nutrient intake and demographic characteristics.

As a conservative approach was taken to making changes to 24-HRs, unusually large portions were changed more often than small portion. For example, 4 kg of watercress is implausible whereas 3 g carrots, may seem unlikely in the context presented, but is entirely plausible. Some of the unusually large portions recorded in self-completed 24-HRs resulted from a lack of detail being presented, for example, a 500 g portion of ice cream being described as ‘as bought’ and the meaning of an ‘average serving’ of pizza or sandwich being open to interpretation. While interviewers were familiar with approximate food weights in grams and used these to guide choices, participants may have little knowledge of food weights in grams. Some of the problems identified could be reduced by optimizing search functions and including additional detail about items and portions, but other issues are more difficult to address.

While time and labour costs involved in interviewer-administered 24-HRs are easy to recognize those for automated systems may not be fully appreciated [28,34]. A full nutrient profile is generated even if a participant records a single food item, providing an illusion of data integrity. However, producing an accurate and valid dataset can be labor intensive as comprehensive quality control checks are required, including identifying outliers and systematically rejecting or correcting implausible responses [18,20].

The main limitation of this analysis is that the study was not designed to compare self-completed and interviewer-administered 24-HRs, therefore participant’s motivation for moving from one method to the other were not routinely recorded. Future research could use qualitative methods to explore the issues relating to non-completion of dietary assessments, but sensitivity would be needed as participants may be reluctant to discuss issues such as poor literacy. Further, participants without an email address were assumed to be unable to self-complete the 24-HR but it is recognized that if it had been possible to deliver a link to them via an alternative method, some may have been capable of self-completing. As the ASCOT research protocol provided an alternative to those unable to self-complete 24-HRs, it allowed us to collect data from individuals who were keen to take part but would otherwise have been excluded from health research. As participants were cancer survivors, they may be more aware of their diet than other individuals. However, the associations between demographic factors and response rates, and issues with self-completion are likely to be relevant to dietary assessment of older adults in general, although applicability to younger individuals may be limited. Participants had expressed an interest in taking part in a lifestyle intervention, which may impact generalizability of results to individuals who did not choose to participate.

## 5. Conclusions

Dietary assessment using self-completed 24-HRs without the option of interviewer-administration may inadvertently contribute to sampling bias and reduce participation by adults who are older, non-white and without formal education qualifications. There is increasing recognition of the need to embed an inequalities perspective in health research and optimizing 24-HR tools and offering interviewer-administered 24-HRs could increase participation by individuals belonging to groups who are often under-served. As dietary surveillance data is used for planning services and policy, the cost advantages of using online self-completed 24-HRs need to be weighed up against reduced response rates and sampling bias.

## Figures and Tables

**Table 1 nutrients-14-05236-t001:** Demographic data, clinical characteristics, internet and email use of cancer survivors, by 24 h dietary recalls (24-HRs) completion method (self-completed and interviewer-administered).

Variable	Total *n* = 1224	Self-Completed 24-HR (*n* = 809, 66.1%)	Interviewer-Administered 24-HR (*n* = 415, 33.9%)	*p*
Age in years (mean, SD)	64.4 (11.3)	61.6 (11.2)	69.8 (9.5)	**<0.001**
Missing data (*n*, %)	3 (0.2)			
Gender (*n*, %)				**<0.001**
Male	465 (38)	279 (34.5)	186 (44.8)	
Female	759 (62)	530 (65.5)	229 (55.2)	
Missing data	0			
Ethnicity (*n*, %)				0.148
White	1136 (92.8)	759 (93.9)	377 (91.7)	
Non-white	83 (6.8)	49 (6.1)	34 (8.3)	
Missing data	5 (0.4)			
Cancer type (*n*, %)				**<0.001**
Breast	667 (54.5)	473 (58.5)	194 (46.7)	
Prostate	320 (26.1)	189 (23.4)	131 (31.6)	
Colorectal	237 (19.4)	147 (18.2)	90 (21.7)	
Missing	0			
Highest level of education (*n*, %)				**<0.001**
No formal qualifications	203 (16.6)	74 (9.6)	129 (34.3)	
GCSE/Vocational	374 (30.6)	255 (33.2)	119 (31.6)	
A-level	162 (13.2)	116 (15.1)	46 (12.2)	
Degree or higher	406 (33.2)	324 (42.1)	82 (21.8)	
Missing data	79 (6.5)			
Marital status (*n*, %)				**<0.001**
Married/cohabiting	872 (71.3)	604 (74.7)	268 (64.7)	
Separated/divorced/widowed	248 (20.3)	134 (54.0)	114 (46.0)	
Single	103 (8.4)	71 (8.8)	32 (7.7)	
Missing data	1 (0.1)			
Employment situation (*n*, %)				**<0.001**
Employed or self-employed	458 (37.4)	363 (45)	95 (15.1)	
Retired	657 (53.7)	373 (46.3)	284 (69.1)	
Other	102 (8.3)	70 (8.7)	32 (7.8)	
Missing data	7 (0.6)			
Index of Multiple Deprivation (IMD) decile (mean, SD)	6.46 (2.5)	6.56 (2.5)	6.27 (2.5)	0.06
Missing data (*n*, %)	63 (5.1)			
BMI category (*n*, %)				0.252
Under/normal weight	428 (35)	287 (37.2)	141 (36.3)	
Overweight	500 (40.8)	341 (44.2)	159 (41)	
Obese	232 (19)	144 (18.7)	88 (22.7)	
Missing data	64 (5.2)			
Internet user (*n*, %)				**<0.001**
Yes	991 (80.1)	752 (93.9)	239 (58.9)	
No	216 (17.6)	49 (6.1)	167 (41.1)	
Missing data	17 (1.4)			
Email user (*n*, %)				**<0.001**
Yes	1036 (84.6)	789 (97.8)	247 (60.2)	
No	181 (14.8)	18 (2.2)	163 (39.8)	
Missing data	7 (0.6)			

Abbreviations: BMI, Body Mass Index; GCSE, General Certificate of Secondary Education; SD, standard deviation. The *p* values below 0.05 are in boldface.

**Table 2 nutrients-14-05236-t002:** Logistic regression analyses for method of completing 24-HR (interviewer-completed 24-HR as target group) (*n* = 1224).

Variable		Unadjusted			Adjusted ^a^	
	OR	95% CI	*p*	OR	95% CI	*p*
Age (years)	1.08	1.07–1.10	**<0.001**	1.09	1.06-1.11	**<0.001**
Gender Male Female	1.00 0.65	- 0.51–0.83	- <0.001	1.00 1.22	- 0.90–1.65	- 0.194
Ethnicity White Non-White	1.00 1.40	- 0.89–2.20	- 0.149	1.00 3.43	- 1.99–5.91	- **<0.001**
Highest Level of Education Degree or higher A-levels GCSE/Vocational No formal qualifications	1.00 1.54 1.79 6.34	- 1.03–2.30 1.28–2.44 4.34–8.90	- **0.035** **<0.001** **<0.001**	1.00 1.84 2.07 4.29	- 1.19–2.85 1.45–2.95 2.87–6.40	- **0.006** **<0.001** **<0.001**
Marital status Married/cohabiting Divorced/Separated/Widowed Single	1.00 1.91 1.01	- 1.43–2.55 0.65-1.52	- **<0.001** 0.958	1.00 1.45 1.25	- 0.84–2.50 0.86–1.83	- 0.178 0.240
Employment Employed Retired Other	1.00 2.90 1.73	- 2.20-3.82 1.07–2.80	- **<0.001** **0.024**	1.00 1.33 1.38	- 0.95–1.88 0.83–2.29	0.099 0.218
IMD (decile)	0.95	0.91–1.00	**0.049**	0.97	0.91–1.03	0.270
BMI Underweight/normal weight Overweight Obesity	1.00 0.97 1.22	- 0.73–1.28 0.87–1.70	- 0.814 0.247	1.00 0.83 1.08	- 0.61–1.14 0.74–1.56	- 0.250 0.703

Abbreviations: CI, confidence interval; OR, odds ratio; IMD, Index of Multiple Deprivation; BMI, Body Mass Index. ^a^ Adjusted for Age, Gender, Ethnicity, Highest level of Education, Marital status, Employment status, IMD, BMI category. The *p* values below 0.05 are in boldface.

**Table 3 nutrients-14-05236-t003:** Energy and nutrient intake and number of items reported, by 24-HR completion method (self-completed and interviewer-administered).

Variable	Total (*n* = 1224) Mean (SD)	Self-Completed 24-HR (*n* = 809) Mean (SD)	Interviewer-Administered 24-HR (*n* = 415) Mean (SD)	*p*
Energy (kcals)	1697 (569)	1661 (566)	1768 (568)	**0.002**
Energy (kJ)	7133 (2383)	6967 (2371)	7430 (2378)	**0.001**
Protein (g)	70.7 (26.9)	69.5 (27.2)	73 (26.3)	**0.033**
Protein (% energy)	17.2 (5.4)	17.2 (5.5)	17.1 (5.1)	0.572
Carbohydrate (g)	194.4 (73.7)	191.2 (74.6)	200.8 (71.6)	**0.030**
Carbohydrate (% energy)	43.4 (9.6)	43.6 (9.9)	43.0 (8.9)	0.274
Total sugar (g)	84.5 (40.5)	81.3 (39.8)	90.6 (41.2)	**<0.001**
Total sugar (% energy)	18.9 (7.4)	18.6 (7.4)	19.5 (7.2)	**0.034**
Fat (g)	67.3 (31.4)	65.1 (31)	71.7 (31.9)	**<0.001**
Fat (% energy)	35.1 (9.2)	34.7 (9.4)	35.9 (8.7)	**0.036**
Saturated fat (g)	24.2 (13.5)	22.7 (12.7)	27.0 (14.6)	**<0.001**
Saturated fat (% energy)	12.5 (4.7)	12.1 (4.5)	13.4 (4.9)	**<0.001**
Fibre (g)	19.5 (9.3)	19.7 (9.8)	19.3 (8.3)	0.551
Number of items	19.72 (7.4)	17.6 (6.2)	23.9 (7.9)	**<0.001**

Abbreviations: SD, standard deviation. The *p* values below 0.05 are in boldface.

**Table 4 nutrients-14-05236-t004:** Prevalence of 24-HRs containing unusually large portions, small portions and items likely to have been selected in error, by 24-HR completion method (self-completed and interviewer administered).

Variable	Total (*n* = 1224) *n* (%)	Self-Completed 24-HR (*n* = 809) *n* (%)	Interviewer-Administered 24-HR (*n* = 415) *n* (%)	*p*
Unusually large portions				
Identified	45 (3.7)	39 (4.8)	6 (1.4)	**0.003**
Changed ^a^	22 (1.8)	19 (2.3)	3 (0.7)	**0.043**
Unusually small portions				
Identified	16 (1.3)	14 (1.7)	1 (0.2)	**0.025**
Changed ^b^	11 (0.9)	10 (1.2)	1 (0.2)	0.081
Probable selection errors				
Identified	41 (3.3)	35 (4.3)	6 (1.4)	**0.008**
Changed ^c^	12 (1)	10 (1.2)	2 (0.5)	0.205

^a^ Examples of large portions changed: watercress 4000 g → 40 g; new potatoes 7 portions (1225 g) → 7 (280 g); ice cream 500 g ‘as bought’ → 124 g (2 portions); brazil nuts 4 ‘average portions’ (100 g) → 4 nuts (15 g). ^b^ Examples of small portions changed: toast: 0 g → 44 g (1 slice); biscuit 1 g → 17 g (1 biscuit); avocado 0.25 g → 35 g (0.25 avocados). ^c^ Examples of probable selection errors changed: dried mushrooms 100 g → raw mushrooms 100 g, oats 350 g → porridge 350 g, chicken skin 400 g → chicken meat & skin 400 g. The *p* values below 0.05 are in boldface.

## Data Availability

Data described in the manuscript will be made available pending application to Rebecca Beeken.

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
