# Peer review of "Comparison between Self-Completed and Interviewer-Administered 24-Hour Dietary Recalls in Cancer Survivors: Sampling Bias and Differential Reporting"

_nutrients, 2022, doi:10.3390/nu14245236_

Round 1
Reviewer 1 Report
Review report
Summary
The paper aims to 1) identify factors associated with being unable to self-complete a 24-HR but able to do so with an interviewer, and 2) explore how dietary data collected by self-completed 24-HR compared to interviewer-administered 24-HRs, in cancer survivors. The aims are clear and focused. The topic of the paper is very important in current times when the population is aging, response-rate in population-based studies is decreasing and the number of tools used to collect dietary data is increasing. As such, the paper can be important for planning of upcoming studies collecting dietary data from populations.
General concept comments
Although the article highlights a clear and important issue, I do have some comment in regards to the generalizability and epidemiological aspect of the study that I would hope had been given more attention.
The participants in the study are cancer survivors, and the authors state that “"Participants were cancer survivors, and while results are likely to be relevant to dietary assessment of older adults in general, applicability to younger individuals may be limited”. However, I would believe that cancer survivors are not directly applicable to the general population in that they might be more aware about their eating habits than participants not experiencing disease previously. If that is the case – than how would that affect the generalizability of the study?
Further, the participants in the current study were, to my understanding, asked to provide contact details if they were interested in attending the study. How would this affect generalizability? Non-responders are often different from responders.
I miss thorough information on how the self-completing group and interview group were categorized. With the information I have now I could not replicate the study.
The authors state that “On the allotted date all participants supplying an email address were sent an email invitation with instructions for self-completing their 24-HR and a personalized electronic link to myfood24 (24). Participants who did not use email were contacted by telephone on the allotted day and a researcher collected dietary information and entered this directly into myfood24”. Does this mean that if they had an email or not allocated them into different groups? Meaning that those with an email address were in the self-complete 24-HR group (unless they were contacted by telephone later if they did not complete), while those not having an email address automatically were in the interviewer-administered 24-HR group? If so – then one would image that those not having an email address are those that are less familiar with technical equipment and thus less likely to answering an online questionnaire? Also, if that is how they were “grouped” I don’t really agree that they are unable to complete the self-reported 24-HR, they really did not have the opportunity to do so if they did not have an email address. I see from table 1 that 60.2% of the interviewed group are email users, but the description in the text is not clear to me at all. Nor is it unclear how the interviews were conducted.
Also –the authors state something about randomization. How and why were they randomized?
The authors mix the use of mode of completion and method of completion – I would stick to one term (for readability).
Finally, to my understanding, those self-completing 24-HR also received some type of support if necessary (could contact the research team if they had questions) – how would this affect the difference between the groups? I’m thinking that large population studies inviting more than 20 000 participants might have challenges with offering this type of support to all. Thus, the differences between the two groups in the current study might be even larger if they did not receive any type of support, which would affect the use of self-completed 24-HR in large studies?
Specific comments
Sentence 12-13: “being unable” and “willing to do so” - are they connected? Are they unable or did they “just” experience challenges when completing it?
Sentence 83-84: I would believe that this impacts the generalizability in regards to how participants differ from non-participants.
Sentence 85: what are the eligibility criteria?
Sentence 87-89: I did not understand this. They were not required to complete recalls at baseline (s. 87) but they were included if they completed the first 24-HR at baseline (s.89-90). Also - randomization must be described if it was performed in the current study.
Sentence 92-93: Would also be interesting to know whether these differed in regards to how we should prepare upcoming surveys (online or paper versions). Just a suggestion.
Sentence 98: GCSE?
Sentence 104: “healthy weight” or “normal weight”?
Sentence 111-113: “Letters were sent to participants informing them of the dates they were due to complete 24-HRs, which were scheduled to collect dietary data for one weekday and one weekend day”. It is unclear which of these measures/days that were used in the results, or if both days are included in the results?
Sentence 115-116: “Participants who did not use email were contacted by telephone on the allotted day and a researcher collected dietary information and entered 116 this directly into myfood24.” Does this mean that they were allocated into self-report and interview-group at this stage?
Sentence 142: RC, VI, SE, GH? If these are person initials, are they necessary in the text?
Sentence 154: Please check if myfood24 should be written “myfood24®”.
Sentence 161: Is 3.75 kcal/g correct or should it be 4 kcal/g as for protein?
Sentence 164-165: Should specify the statistical package, “(IBM Corp., Armonk, N.Y., USA)”
Sentence 171: Add the independent variables in parenthesis (to make it clear)
Sentence 172: “.. one logistic regression analysis/model was run…”
Sentence 178-179: I agree, and I would also believe that this is an important aspect if the two groups are grouped according to having an email address or not having an email address.
Sentence 185: I would like a sentence after this, concluding to whether imputation changed the results compared to the not-imputed sample.
Sentence 207-209: Good that this is described.
Sentence 242: “…24-HRs increasingly being the method of”?
Sentence 243: Irrespective or research protocol or research method?
Sentence 244: “…reduce response rates and alter the results”. I suggest this change if the authors agree with the statement.
Sentence 257: Ascot à ASCOT
Sentence 276: I’m not familiar with presenting odds as percentage. But if doing so wouldn’t 4.29 OR be 326%? Considering that an OR of 1 is 0, 2 =100%, 3=200%, 4=300% and so on?
Sentence 311-314: The nutrient intakes are different in the two groups, but this could also be actual differences as the groups also are different according to other characteristics – not only due to underreporting or the method used. I would add a sentence considered that as an option as well.
Table 2:
- I suppose formatting of tables will be done later but try keeping descriptions in one line (more readable).
- “Other” under “employment” is skewed.
- “Marital status” - “Single” round the CI (1.517à1.52)?
- The variables and their description (i.e. what level that is used as a reference category) should be described in the statistical analysis section and as footnote in the table.
- CI should be described as 95% CI (if that is what is being used)
Table 3:
- I suppose formatting of tables will be done later but try keeping descriptions in one line (more readable).
- Should “KJ” be “kJ”?
Table 4:
- “Examples of changes in the text” – could this rather be presented as footnote? Or elsewhere?
Author Response
Dear Reviewer,
We would like to thank you for taking the time to review our manuscript (ISSN 2072-6643) and for providing insightful and useful comments. We have now addressed these with a view to strengthening the paper.
Point-by-point responses are provided below.
Review report
Summary
The paper aims to 1) identify factors associated with being unable to self-complete a 24-HR but able to do so with an interviewer, and 2) explore how dietary data collected by self-completed 24-HR compared to interviewer-administered 24-HRs, in cancer survivors. The aims are clear and focused. The topic of the paper is very important in current times when the population is aging, response-rate in population-based studies is decreasing and the number of tools used to collect dietary data is increasing. As such, the paper can be important for planning of upcoming studies collecting dietary data from populations.
General concept comments
Although the article highlights a clear and important issue, I do have some comment in regards to the generalizability and epidemiological aspect of the study that I would hope had been given more attention.
The participants in the study are cancer survivors, and the authors state that “"Participants were cancer survivors, and while results are likely to be relevant to dietary assessment of older adults in general, applicability to younger individuals may be limited”. However, I would believe that cancer survivors are not directly applicable to the general population in that they might be more aware about their eating habits than participants not experiencing disease previously. If that is the case – than how would that affect the generalizability of the study?
Thank you for raising this issue. We could not find any evidence that cancer survivors are more aware of their diet but agree this could be the case. Lines 365-367 have been edited to read ‘As participants were cancer survivors, they may be more aware of their diet than other individuals. However, the associations between demographic factors and response rates and issues with self-completion are likely to be relevant to dietary assessment of older adults in general, although applicability to younger individuals may be limited.’
Further, the participants in the current study were, to my understanding, asked to provide contact details if they were interested in attending the study. How would this affect generalizability? Non-responders are often different from responders.
We agree that this is an issue with all dietary surveys, but believe this is outside the scope of this paper, which sought to explore completion of 24-HRs among participants who had already consented to take part.
I miss thorough information on how the self-completing group and interview group were categorized. With the information I have now I could not replicate the study.
The authors state that “On the allotted date all participants supplying an email address were sent an email invitation with instructions for self-completing their 24-HR and a personalized electronic link to myfood24 (24). Participants who did not use email were contacted by telephone on the allotted day and a researcher collected dietary information and entered this directly into myfood24”. Does this mean that if they had an email or not allocated them into different groups? Meaning that those with an email address were in the self-complete 24-HR group (unless they were contacted by telephone later if they did not complete), while those not having an email address automatically were in the interviewer-administered 24-HR group? If so – then one would image that those not having an email address are those that are less familiar with technical equipment and thus less likely to answering an online questionnaire? Also, if that is how they were “grouped” I don’t really agree that they are unable to complete the self-reported 24-HR, they really did not have the opportunity to do so if they did not have an email address. I see from table 1 that 60.2% of the interviewed group are email users, but the description in the text is not clear to me at all. Nor is it unclear how the interviews were conducted.
Thank you, we appreciate the need for clarification here. The following sentences have been added to the methods section ‘Participants without an email address were therefore automatically allocated to interviewer-administered 24-HRs.’ (line 119-121). ‘Participants were therefore defined as being ‘unable’ to self-complete a 24-HR if they didn’t have an email address or if they reported experiencing considerable problems when attempting self-completion. (line 131-133). A sentence has also been added to the limitations section ‘Further, participants without an email address were assumed to be unable to self-complete the 24-HR but it is recognized that if an electronic link had been delivered to them via an alternative method, some may have been capable of self-completing. (line 359-361)
Also –the authors state something about randomization. How and why were they randomized?
This refers to randomization to ASCOT but this was unclear and not relevant to the current analysis so has been deleted (line 88-90).
The authors mix the use of mode of completion and method of completion – I would stick to one term (for readability).
This is a useful observation and we have now used ‘method’ of completion throughout.
Finally, to my understanding, those self-completing 24-HR also received some type of support if necessary (could contact the research team if they had questions) – how would this affect the difference between the groups? I’m thinking that large population studies inviting more than 20 000 participants might have challenges with offering this type of support to all. Thus, the differences between the two groups in the current study might be even larger if they did not receive any type of support, which would affect the use of self-completed 24-HR in large studies?
This is an insightful comment that we had not considered. The following sentence has been added to the discussion ‘In large surveys a telephone help lines or other methods of support may not be provided, which could exacerbate sampling bias and problems in self-completing 24-HRs.’ (line 274-276)
Specific comments
Sentence 12-13: “being unable” and “willing to do so” - are they connected? Are they unable or did they “just” experience challenges when completing it?
We debated this wording ourselves, if telephone interviews had not been available some participants may have found a way to self-complete with help from family or others but, as described in the methods, those not initially self-completing and then telling us they were struggling, were offered a phone interview. The wording here is to make it clear that these individuals weren’t dropping out because of lack of motivation/being too busy. For clarity we have added the following definition to the methods ‘Participants were therefore defined as being ‘unable’ to self-complete a 24-HR if they didn’t have an email address or if they reported experiencing considerable problems when attempting self-completion.’ (line 131-133).
Sentence 83-84: I would believe that this impacts the generalizability in regards to how participants differ from non-participants.
Thank you for highlighting this issue. The following sentence has been added to the limitations ‘Participants had expressed an interest in taking part in a lifestyle intervention, which may impact generalizability of results to individuals who did not choose to participate.’ (line 368-370).
Sentence 85: what are the eligibility criteria?
The eligibility criteria were described earlier in the paragraph, ‘outlined above’ has been added to this sentence for clarity.
Sentence 87-89: I did not understand this. They were not required to complete recalls at baseline (s. 87) but they were included if they completed the first 24-HR at baseline (s.89-90). Also - randomization must be described if it was performed in the current study.
We can see this line could cause confusion and have deleted it.
Sentence 92-93: Would also be interesting to know whether these differed in regards to how we should prepare upcoming surveys (online or paper versions). Just a suggestion.
This is a good idea and certainly something we could look into further.
Sentence 98: GCSE?
This is General Certificate of Secondary Education, and has now been written out in full.
Sentence 104: “healthy weight” or “normal weight”?
Thank you for highlighting, this should read ‘normal weight’ and has been edited on lines 104 – 105 and in Table1 and Table 2.
Sentence 111-113: “Letters were sent to participants informing them of the dates they were due to complete 24-HRs, which were scheduled to collect dietary data for one weekday and one weekend day”. It is unclear which of these measures/days that were used in the results, or if both days are included in the results?
The first recall was scheduled for a weekday but some participants completed the recall for a weekend day. The methods have been edited to read ‘On the allotted date for the first 24-HR, which was a weekday, all participants supplying an email address were sent an email invitation’ (line 114-115) and the following sentence has been added to the results ‘The first recall was scheduled for a weekday but was sometimes completed on a weekend day.’ (Line 202-203)
Sentence 115-116: “Participants who did not use email were contacted by telephone on the allotted day and a researcher collected dietary information and entered 116 this directly into myfood24.” Does this mean that they were allocated into self-report and interview-group at this stage?
Participants were not allocated to self-report or interviewer group by the research team. Those able and willing to self-complete, did so. Lines 88-90, may have caused confusion, suggesting participants were allocated/randomized and these has now been deleted.
Sentence 142: RC, VI, SE, GH? If these are person initials, are they necessary in the text?
These were initials of authors, which have now been deleted.
Sentence 154: Please check if myfood24 should be written “myfood24®”.
We are glad this was suggested. We have edited the paper throughout to read myfood24®and also ASA24®.
Sentence 161: Is 3.75 kcal/g correct or should it be 4 kcal/g as for protein?
This was double checked in the UK Food Tables (McCance and Widdowson) and 3.57kcal/g is correct.
Sentence 164-165: Should specify the statistical package, “(IBM Corp., Armonk, N.Y., USA)”
Thank you, this has been added.
Sentence 171: Add the independent variables in parenthesis (to make it clear)
This has been added as suggested for clarity. (line177)
Sentence 172: “.. one logistic regression analysis/model was run…”
The word ‘analysis’ has been added to this sentence.
Sentence 178-179: I agree, and I would also believe that this is an important aspect if the two groups are grouped according to having an email address or not having an email address.
We agree and did not include email use in the regression model.
Sentence 185: I would like a sentence after this, concluding to whether imputation changed the results compared to the not-imputed sample.
We have included this detail in the results (Line 217-218).
Sentence 207-209: Good that this is described.
Sentence 242: “…24-HRs increasingly being the method of”?
This has been added.
Sentence 243: Irrespective or research protocol or research method?
This has been changed from ‘protocol’ to ‘method’ for clarity.
Sentence 244: “…reduce response rates and alter the results”. I suggest this change if the authors agree with the statement.
Thank you – the sentence has been edited as suggested.
Sentence 257: Ascot à ASCOT
This has been changed to ASCOT, for consistency throughout the paper.
Sentence 276: I’m not familiar with presenting odds as percentage. But if doing so wouldn’t 4.29 OR be 326%? Considering that an OR of 1 is 0, 2 =100%, 3=200%, 4=300% and so on?
Rather than presenting the odds as a percentage we have edited this sentence to ‘For adults without formal educational qualifications, the odds of being unable to self-complete a 24-HR are 4.29 (CI 2.87-6.40) times those of individuals with a degree. (line294-296).
Sentence 311-314: The nutrient intakes are different in the two groups, but this could also be actual differences as the groups also are different according to other characteristics – not only due to underreporting or the method used. I would add a sentence considered that as an option as well.
This is a sensible point to consider and the following sentence has been added ‘This could be due to the different characteristics of the two groups but other factors may also explain the discrepancy.’ (Line 320-321)
Table 2:
- I suppose formatting of tables will be done later but try keeping descriptions in one line (more readable).
- “Other” under “employment” is skewed.
- “Marital status” - “Single” round the CI (1.517à1.52)?
This has been rounded as suggested.
- The variables and their description (i.e. what level that is used as a reference category) should be described in the statistical analysis section and as footnote in the table.
The following sentence has been added to the statistical analysis section ‘Reference categories were male, white, degree or higher, married/co-habiting, employed and underweight/healthy weight.’ (Line180-182). A footnote wasn’t added to Table 2 as we felt it was clear that the first option for each variable was the reference category as the OR was 1.00 and not CI was presented.
- CI should be described as 95% CI (if that is what is being used)
It is the 95% CI that is presented so this has been added to Table 2.
Table 3:
- I suppose formatting of tables will be done later but try keeping descriptions in one line (more readable).
- Should “KJ” be “kJ”?
Thank you, this has been changed to kJ.
Table 4:
- “Examples of changes in the text” – could this rather be presented as footnote? Or elsewhere?
We appreciate this suggestion and have moved the examples to a footnote.

Reviewer 2 Report
Diet misreporting through self-report instruments is well known. Because of changes in lifestyles, because of globalization, the evaluation of diet as a risk factor for chronic degenerative problems has drawn the attention of researchers in recent years, due to the feasibility of being modifiable. Therefore, it is of great importance to have an accurate and reliable dietary assessment since misinformation about diet can lead to erroneous estimates regarding diet.
Hence, the topic of the manuscript is very important, having a greater and better understanding of the limitations of dietary measurement instruments will allow us to know the scope of our results. Congratulations to the authors, the manuscript seemed very clear to me, the authors explain all the procedures implemented, the biases they faced and how they handled them.
I only have one doubt, they used some method for classifying misreporters of energy and nutrients intake between the modes of dietary recall completion.
I only have two minor recommendations, review the second column of table 1, a box can be seen in the title. Likewise, in table 1, in the age row, leave a single decimal like the rest of the table.
From then on, I have no further comments.
Author Response
Dear Reviewer,
We would like to thank you for taking the time to review our manuscript (ISSN 2072-6643) and for providing insightful and useful comments.
Comments and Suggestions for Authors
Diet misreporting through self-report instruments is well known. Because of changes in lifestyles, because of globalization, the evaluation of diet as a risk factor for chronic degenerative problems has drawn the attention of researchers in recent years, due to the feasibility of being modifiable. Therefore, it is of great importance to have an accurate and reliable dietary assessment since misinformation about diet can lead to erroneous estimates regarding diet.
Hence, the topic of the manuscript is very important, having a greater and better understanding of the limitations of dietary measurement instruments will allow us to know the scope of our results. Congratulations to the authors, the manuscript seemed very clear to me, the authors explain all the procedures implemented, the biases they faced and how they handled them.
I only have one doubt, they used some method for classifying misreporters of energy and nutrients intake between the modes of dietary recall completion.
Thank you for this complementary review. We agree these are important issues and hope our paper contributes to a better understanding of the limitations of dietary assessment. No attempt was made to identify misreporters of energy or nutrients in this paper, but it is something that we will address when using the full dataset to evaluate the ASCOT intervention.
I only have two minor recommendations, review the second column of table 1, a box can be seen in the title. Likewise, in table 1, in the age row, leave a single decimal like the rest of the table.
These two recommendations have been followed and Table 1 has been edited accordingly.
From then on, I have no further comments.